# Age-Driven Genetic and Epigenetic Heterogeneity in B-ALL

**DOI:** 10.3390/ijms26188774

**Published:** 2025-09-09

**Authors:** Yoana Veselinova, Manel Esteller, Gerardo Ferrer

**Affiliations:** 1Cancer Epigenetics, Josep Carreras Leukaemia Research Institute (IJC), 08916 Badalona, Spain; 2Institució Catalana de Recerca i Estudis Avançats (ICREA) and Universitat Pompeu Fabra, 08010 Barcelona, Spain; 3Physiological Sciences Department, School of Medicine and Health Sciences, University of Barcelona (UB), 08036 Barcelona, Spain; 4Centro de Investigación Biomédica en Red Cáncer (CIBERONC), 28029 Madrid, Spain; 5Institut d’Investigació en Ciències de la Salut Germans Trias i Pujol, 08916 Badalona, Spain; 6Karches Center for Oncology Research, The Feinstein Institutes for Medical Research, Northwell Health, Manhasset, NY 11030, USA

**Keywords:** B-cell acute lymphoblastic leukemia, genetics, epigenetics, pediatric, young adult, adult

## Abstract

B-cell acute lymphoblastic leukemia (B-ALL) remains a major clinical challenge in hematologic oncology, characterized by a continuous evolution of molecular drivers that shape its heterogeneity across the age spectrum. Pediatric B-ALL is generally associated with high cure rates, while adult forms of the disease are often more aggressive and less responsive to treatment. This review examines the age-specific genetic and epigenetic landscapes that contribute to this disparity, revealing how the nature and timing of molecular alterations point to fundamentally different leukemogenic processes. Favorable genetic aberrations, such as *ETV6::RUNX1* and hyperdiploidy, are predominant in children, whereas adults more frequently present with high-risk features, including *BCR::ABL1* fusions and *IKZF1* deletions. Epigenetic distinctions are similarly age-dependent, involving divergent patterns of DNA methylation, histone modifications, and non-coding RNA expression. For example, pediatric B-ALL frequently harbors mutations in epigenetic regulators like *SETD2* and *CREBBP*, while adult B-ALL is more commonly affected by alterations in *TET2* and *IDH1/2*. These molecular differences are not only prognostic but also mechanistic, reflecting distinct developmental trajectories and vulnerabilities. Understanding these age-driven transitions is essential for improving risk stratification and developing precision therapies tailored to the unique biology of B-ALL across the lifespan.

## 1. Introduction

B-cell acute lymphoblastic leukemia (B-ALL) is a highly heterogeneous hematologic malignancy characterized by the uncontrolled proliferation of immature B-cell precursors, leading to bone marrow (BM) failure and systemic disease [1]. These malignant B-cell precursors often harbor genetic and epigenetic alterations that drive unchecked proliferation, block normal differentiation, and promote leukemic cell survival [2,3,4]. Although B-ALL can develop at any age, it predominantly affects children, with a peak incidence between two and five years of age. In contrast, adult B-ALL is less common but exhibits distinct biological and clinical features that significantly impact prognosis and therapeutic response [5]. While commonly categorized into pediatric and adult forms, accumulating evidence indicates a continuous age-related evolution of molecular drivers, rather than a sharp dichotomy between the two.

Pediatric B-ALL accounts for approximately 80% of childhood leukemia cases and has one of the highest cure rates among hematologic malignancies. While B-ALL is more common in children, adult cases still represent a significant portion, making up approximately 40% of all B-ALL diagnoses [6,7]. Pediatric patients typically present with symptoms related to BM failure, including anemia, thrombocytopenia, and neutropenia, often accompanied by fever, fatigue, and bleeding. Adult patients with B-ALL, on the other hand, frequently exhibit higher leukocyte counts at diagnosis, and a greater proportion present with extramedullary involvement, such as central nervous system (CNS) infiltration, which can complicate treatment. Additionally, adult patients are more likely to have hepatosplenomegaly and lymphadenopathy at presentation, reflecting a more aggressive disease course [8].

The prognosis for pediatric B-ALL has dramatically improved over the past few decades. In the 1960s, the presentation of B-ALL was almost a death sentence; however, in developed countries, long-term survival rates now exceed 85–90%. This success is largely attributed to the implementation of risk-adapted, intensive chemotherapy regimens, effective CNS prophylaxis, and advancements in measurable residual disease (MRD) monitoring [9,10]. In contrast, adult B-ALL remains a therapeutic challenge, with five-year survival rates ranging from 40% to 50%. Adults are more likely to experience treatment-related toxicity, have lower tolerance to intensive chemotherapy, and face higher rates of relapse, necessitating alternative therapeutic strategies [8,11,12].

Standard chemotherapy regimens for pediatric B-ALL are highly effective, incorporating corticosteroids, vincristine, asparaginase, and anthracyclines. These regimens have been adapted for adolescent and young adult (AYA) patients, who tend to have better outcomes when treated with pediatric-inspired protocols rather than conventional adult regimens. However, in older adults, the use of intensive chemotherapy is limited by comorbidities and treatment toxicity. As a result, targeted therapies such as tyrosine kinase inhibitors (TKIs) for Philadelphia-positive (Ph+) B-ALL, monoclonal antibodies like blinatumomab and inotuzumab ozogamicin, and CAR-T cell therapy have become crucial components of adult treatment strategies.

Pediatric and adult B-ALL differ significantly in their genetic landscapes, influencing disease classification, prognosis, and therapeutic response. While pediatric B-ALL is often characterized by genetic alterations associated with favorable outcomes, adult B-ALL more frequently harbors high-risk genetic abnormalities linked to poorer survival rates [5,13]. This review will first provide a comparative analysis of the key genetic differences between pediatric and adult B-ALL, highlighting their implications for disease progression and treatment response. Nevertheless, researchers have found age-related differences in drug response within genetic ALL subtypes [14]. Beyond genetic alterations, increasing evidence underscores the role of epigenetic dysregulation—including DNA methylation, histone modifications, and chromatin remodeling—as a key driver of leukemogenesis. Differences in epigenetic landscapes between pediatric and adult B-ALL may contribute to variations in disease biology, therapeutic resistance, and long-term outcomes. While epigenetic dysregulation is a hallmark of leukemogenesis across ages, epigenetic research and translational progress have historically advanced faster in pediatric B-ALL than in adult disease. Several factors contribute to this discrepancy. First, pediatric B-ALL is more common than its adult counterpart, enabling the assembly of large cooperative cohorts for genomic and epigenomic profiling and adding biological significance to clinical trials. Second, pediatric B-ALL has higher cure rates and more standardized treatment schemes, which facilitate the identification of molecular correlations of therapy response. In contrast, adult B-ALL is biologically more heterogeneous, diagnosed often in individuals with comorbidities, and subdivided into smaller, less uniformly treated cohorts. Additionally, adult cases more frequently relapse early or die of treatment-related toxicity, reducing the availability of longitudinal cases for mechanistic studies. These obstacles, together with a historical focus on the recurrent *BCR::ABL1* and *TP53* lesions, have hindered the integration of comprehensive epigenomic profiling and drug development into adult B-ALL research.

A deeper understanding of the epigenetic differences holds promise for the development of novel therapeutic strategies aimed at reversing epigenetic dysregulation and improving survival across all age groups. Thus, the purpose of this review is to synthesize and appraise current knowledge on genetic and epigenetic heterogeneity of B-ALL in relation to age-associated patterns, with particular focus on pediatric/childhood, AYA, and adult cohorts as defined in major genomic and epigenomic studies. In many datasets, AYAs are combined with either pediatric or adult groups, limiting the resolution of truly age-specific biology, but still reveal important trends that merit discussion. Compared with previous reviews, we place particular focus on: (1) integrating recent large-scale multi-omics studies that resolve age and subtype specific epigenetic alterations, (2) highlighting novel therapeutic targets emerging from these analyses; and (3) discussing how these findings support the development of age-adapted, epigenetically informed treatment strategies. By emphasizing the patterns identified within the available stratifications, our aim is to provide a realistic, evidence-based perspective on how current knowledge can inform both clinical decision-making and future research priorities.

## 2. Genetics and Transcriptomics of Pediatrics vs. Adult B-ALL

B-ALL exhibits a wide range of molecular alterations that contribute to its pathogenesis and heterogeneity, influencing clinical outcomes. These alterations span various layers of genomic architecture, including copy number alterations (CNAs) (e.g., hyperdiploidy and gene duplications), structural chromosomal abnormalities (translocations and gene fusions), point mutations in critical genes, and, more recently, transcriptional signatures that define specific disease subtypes. The type and frequency of these genetic and transcriptomic events vary markedly between pediatric and adult patients, with profound implications for risk stratification, treatment response, and overall prognosis. This section summarizes the main genetic and transcriptomic differences between pediatric and adult B-ALL (Table 1), emphasizing how these molecular features relate to clinical outcomes and have been previously reviewed [1,15,16,17,18,19].

Chromosomal aneuploidy plays a significant prognostic role in B-ALL. High hyperdiploidy, defined as a chromosomal count exceeding 50, is one of the most common cytogenetic abnormalities in childhood B-ALL, present in approximately 25–30% of cases. It is associated with a favorable prognosis due to its sensitivity to chemotherapy. In contrast, hypodiploidy—fewer than 44 chromosomes—is rarer but more frequently observed in adults and confers a poor prognosis due to its association with genetic instability and treatment resistance.

Structural chromosomal abnormalities, particularly translocations and gene fusions, also show age-dependent patterns. The ETV6-RUNX1 fusion (t(12;21)), detected in 20–25% of pediatric B-ALL patients, is rare in adults and is linked to excellent long-term survival in children. Conversely, the BCR-ABL1 fusion (Philadelphia chromosome, t(9;22)), which occurs in only 3% of pediatric B-ALL cases, is significantly more prevalent in adults, accounting for nearly 25% of cases. Philadelphia-positive (Ph+) B-ALL has historically been associated with poor prognosis in adults, though tyrosine kinase inhibitors (TKIs) have improved outcomes substantially. *KMT2A* rearrangements (MLLr), common in infant B-ALL, are also observed in up to 10% of adult cases and are associated with particularly aggressive disease and poor survival rates.

Beyond chromosomal alterations, gene mutations and CNAs further distinguish pediatric from adult B-ALL. *IKZF1* deletions, which impair the function of the Ikaros transcription factor, are present in ~15% of pediatric cases but occur nearly twice as frequently in adults (~30%). These deletions, especially in Ph-like B-ALL, are associated with treatment resistance and worse prognosis. TP53 mutations are another hallmark of adult B-ALL, seen in up to 15% of cases compared to fewer than 5% in children. Their presence confers significant chemoresistance and a high risk of relapse. Deletions of CDKN2A/B, critical regulators of the cell cycle, are found in both age groups but are more prevalent and more strongly linked to poor prognosis in adults. RAS pathway mutations, including those in *NRAS*, *KRAS*, and *PTPN11*, occur in 15–20% of pediatric B-ALL cases and are associated with variable degrees of chemotherapy resistance. Although less frequent in adults, these mutations still contribute to adverse outcomes in some patients. Additionally, mutations in epigenetic regulators such as *CREBBP*, *SETD2*, and *EZH2* are more commonly observed in adult B-ALL cases. These mutations may contribute to disease progression and therapeutic resistance by altering chromatin accessibility and transcriptional control.

Several transcriptional subtypes of B-ALL have emerged as clinically relevant, particularly in cases lacking defining cytogenetic alterations. Philadelphia-like (Ph-like) B-ALL, characterized by a gene expression profile similar to Ph+ B-ALL but lacking the BCR-ABL1 fusion, occurs in 10–15% of pediatric and up to 30% of adult cases, and is associated with poor prognosis, chemotherapy resistance, and frequent *IKZF1* deletions. Other rare subtypes, such as *ETV6*::*RUNX1*-like and KMT2A-like B-ALL, mimic the transcriptional signatures of their respective fusion-positive counterparts but lack the canonical rearrangements. These occur more often in children and involve alternative fusions or CNAs disrupting lymphoid development. Together, these transcriptomic subtypes highlight the importance of gene expression profiling for more precise classification and risk stratification in both pediatric and adult B-ALL.

## 3. Epigenetics of Pediatric vs. Adult B-ALL

Epigenetic modifications represent a fundamental layer of gene regulation that does not involve changes to the underlying DNA sequence. Instead, they modulate how genetic information is interpreted and utilized by cells, often through chemical changes that are dynamic and reversible. These include DNA methylation, histone post-translational modifications (PTMs), 3D chromatin organization, and non-coding RNAs (ncRNAs), which are introduced by enzymatic “writers”, interpreted by “readers” and “remodelers”, and removed by “erasers” [20]. This regulation is highly plastic and sensitive to environmental and behavioral influences such as toxins, drugs, diet, and stress, as well as to the cellular microenvironment, including cytokines and intercellular signals. Epigenetic control is essential for diverse biological processes, including transcription, DNA repair, and differentiation. In the immune system, it governs the commitment of hematopoietic stem cells to mature immune lineages and orchestrates critical functions such as immunoglobulin gene rearrangement, B cell maturation, T helper cell differentiation, and cytokine production. In cancer, epigenetic alterations are frequent, often precede genetic mutations, and contribute to tumor initiation, progression, and resistance. These changes are now recognized as a hallmark of cancer [21]. Given that the development of the hematopoietic system relies heavily on epigenetic regulation, it is not surprising that most cases of hematopoietic malignancies, including B-ALL, exhibit some form of epigenetic dysregulation.

### 3.1. Genetic Mutations in Epigenetic Regulators

Mutations in genes encoding epigenetic regulators are increasingly recognized as key contributors to B-ALL pathogenesis, with distinct patterns emerging across age groups and disease subtypes. In pediatric B-ALL, *SETD2* loss-of-function mutations were first reported by Mar et al., occurring in approximately 12% of cases at diagnosis, particularly enriched in MLLr and *ETV6::RUNX1* subtypes [22]. These mutations often co-occurred with RAS pathway alterations and were found to increase in frequency and diversity at relapse, alongside mutations in *CREBBP*, *MSH6*, *KDM6A*, and *KMT2B*. Similarly, Brady et al. highlighted that DUX4r pediatric B-ALL exhibited the highest prevalence of epigenetic alterations (66%), predominantly affecting *KMT2D*, *TBL1XR1*, and *SETD2*, while hyperdiploid and near-haploid subtypes frequently harbored *CREBBP* mutations [19]. In contrast, adult and AYA B-ALL cases show a broader and more frequent spectrum of epigenetic mutations. Xiao et al. demonstrated that in Ph-negative adult B-ALL, mutations in epigenetic regulators increased from 16.7% at diagnosis to 46.4% at relapse post-transplant, with *SETD2*, *CREBBP*, *KDM6A*, and *NR3C1* being the most affected [23]. Ou et al. further characterized these alterations in a large AYA/adult cohort, identifying mutations in *TET2*, *KMT2A*, and *PHF6* in 31.4% of B-ALL patients and associating them with poorer outcomes, particularly in cytogenetically normal cases [24]. Notably, mutations in epigenetic modifiers such as *KMT2B*, *SETD2*, *ASXL1*, *EZH2*, *TET2*, and *KDM5C* are more frequent in adults than in children, especially in the MLLr subtype [25].

A particularly high-risk subgroup in AYA/adult B-ALL is defined by *IDH1/2* mutations, which lead to the accumulation of the oncometabolite 2-hydroxyglutarate (2-HG). This metabolite inhibits *TET2* and histone demethylation, resulting in widespread DNA and histone methylation [26,27]. Yasuda et al. identified this *IDH1/2*-mutant subtype as clinically distinct and more prevalent in AYA/adults than in children, despite presenting normal WBC counts. They also confirmed that low-hypodiploid cases consistently harbor *CREBBP* heterozygous loss, echoing findings in pediatric cohorts [28].

Collectively, these findings underscore the age- and subtype-specific landscape of epigenetic mutations in B-ALL. Pediatric cases tend to harbor mutations in *SETD2*, *CREBBP*, and *KMT2D*, often in association with specific cytogenetic subtypes, while AYA and adult cases exhibit a broader range of alterations, including *TET2*, *IDH1/2*, and *PHF6*, with significant implications for prognosis and therapeutic stratification.

### 3.2. DNA Methylation

#### 3.2.1. Specific Gene Promoter Methylation

Early studies on DNA methylation in B-ALL began with Leegwater et al. in 1997, who analyzed the promoter region of the *CALCA* gene in pediatric ALL patients. They found increased CpG methylation in nearly all cases at diagnosis compared to healthy donors, and in most of the patients followed through relapse, methylation levels were even higher, suggesting a progressive epigenetic silencing during disease evolution [29]. In 1999, P.G. Corn et al. found the methylation of *TP73* promoter resulting in its silencing in ALL [30]. Later, García-Manero et al. expanded the scope by analyzing the methylation status of ten gene promoters in adult ALL, including *ABCB1*, *THBS2*, *MYOD1*, *ESR1*, *CDKN2B*, *THBS1*, *MME*, *ABL1*, *CDKN2A*, and *TP73*. They found that 86.2% of cases had at least one methylated gene, with *THBS2*, *THBS1*, *MYOD1*, and *ESR1* showing the highest methylation densities in B-ALL. Interestingly, *ABCB1* and *THBS1* methylation were more frequent in Ph-negative cases, and only *THBS1* methylation was associated with favorable clinical outcomes [31].

Promoter methylation of cell cycle regulators such as *CDKN1C* (p57), *CDKN2A* (p16), *CDKN2B* (p15), and *TP73* has been extensively studied in AYA and adult B-ALL. Shen et al. reported that p57 methylation occurred in 50% of cases and was associated with transcriptional silencing, which was reversible with azacitidine treatment. Ph-negative patients with methylation in two or more of these genes had significantly worse overall survival compared to those with none or only one methylated gene [32]. Bueso-Ramos et al. confirmed these findings and showed that while p73 was the only gene with a clear methylation-expression correlation, expression of p57, p15, or their combinations were associated with better survival outcomes [33]. In pediatric B-ALL, the same genes showed a markedly different methylation profile. Only 7% of cases had *CDKN1C* promoter methylation, and just 3% of Ph-negative patients showed methylation of *TP73*, *CDKN2B*, or *TP53*. Despite this, p57 transcript levels were significantly lower than in normal lymphocytes in over half of the cases, suggesting that transcriptional silencing in children may occur through mechanisms other than DNA methylation [34].

Several studies have identified subtype-specific methylation patterns. In MLLr pediatric B-ALL, Stumpel et al. described recurrent hypomethylation of proto-oncogene promoters including *HOXA9*, *PARK7*, *DIAPH1*, *SFMBT1*, *RAN*, *SET*, *RUNX1*, and *MYC*. High expression of *SET*, *RUNX1*, *MYC*, and *SFMBT1* was associated with increased relapse risk. Treatment with HDAC inhibitors led to degradation of the MLL-AF4 fusion protein and restoration of promoter methylation, suggesting a therapeutic strategy for this subtype [35].

Geng et al. performed an integrative analysis of promoter methylation and chromatin binding in adult B-ALL subtypes, including *BCR::ABL1*, *E2A::PBX1*, and MLLr. In *BCR::ABL1* cases, they identified hypomethylation and overexpression of *IL2RA* (CD25), which was associated with poor prognosis. In *E2A::PBX1* B-ALL, the fusion protein was found to bind directly to promoter regions of genes such as *CALD1*, *ARL4C*, *ST6GALNAC3*, and *EXTL3*, leading to hypomethylation and overexpression, often in cooperation with the histone acetyltransferase p300. In contrast, genes like *MLKL*, *SERINC5*, and *CTDSP2* were hypermethylated and underexpressed. MLLr cases exhibited a distinct hypomethylation pattern associated with MLL-AF4 binding and H3K79me2 deposition, characterized by the overexpression of genes such as *FLT3* and *BCL6*, the latter of which has been proposed as a therapeutic target [36].

Other genes have also been implicated in B-ALL through promoter methylation. *SLIT2* was found to be methylated in 58% of pediatric B-ALL cases, with expression silenced in cell lines and restored after azacitidine treatment [37]. *CDKN2A*, *CSMD1*, and *COL6A* were found to be hypermethylated at relapse in pediatric and AYA patients, with corresponding downregulation of expression. Treatment with decitabine restored gene expression, suggesting a role for methylation in drug resistance [38,39]. Musialik et al. focused on hematopoietic transcription factors and found aberrant hypermethylation of *TAL1*, *IRF8*, *MEIS1*, and *IRF4* in a subset of pediatric patients. Silencing of *IRF8*, *MEIS1*, and *TAL1* was confirmed at the RNA level, and low *MEIS1* expression was associated with higher WBC counts and poor prognosis, suggesting potential use in risk stratification [40]. The Notch-Hes signaling pathway has also been implicated in pediatric B-ALL. Promoters of *NOTCH3*, *JAG1*, *HES2*, *HES4*, and *HES5* were frequently hypermethylated, with *HES5* showing reduced expression and repressive histone marks. Restoration of *HES5* expression induced apoptosis and cell cycle arrest, supporting its role as a tumor suppressor [41]. In the context of treatment resistance, Abdullah et al. identified *ADAMTSL5*, *CDH11*, and *ZNF214* as differentially methylated in chemoresistant pediatric B-ALL cases. While the functional consequences of these changes remain to be fully elucidated, they may serve as early markers of resistance [42,43,44,45]. Additional studies have highlighted unique epigenetic features in adult B-ALL. Loeff et al. identified a CD52-negative subpopulation at diagnosis, which is absent in other B-cell malignancies and in healthy donors. This population lacked *PIGH* expression, which is responsible for GPI anchor introduction, due to promoter methylation and repressive histone marks, and this deficiency could be reversed by azacitidine. This finding may explain the limited efficacy of alemtuzumab in B-ALL and suggests that epigenetic drugs could enhance therapeutic responses [46]. Ghantous et al. provided compelling evidence for the prenatal epigenetic origin of B-ALL. By analyzing methylation profiles from birth to relapse, they identified *VTRNA2-1* as hypermethylated at birth in children who later developed B-ALL. Its methylation status correlated with expression, disease progression, and outcome, supporting the idea that epigenetic priming may occur in utero [47]. Yasuda et al. identified a Ph-negative high-risk AYA and adult B-ALL subtype characterized by CDX2 overexpression driven by promoter hypomethylation and recurrent 1q gain. CDX2, normally silenced in adult hematopoiesis, modulates HOX expression and may contribute to leukemogenesis in this context [28]. On the other hand, *FHIT* promoter hypermethylation was observed in pediatric and AYA cases and associated with higher WBC counts, although its clinical significance remains to be clarified [48].

These findings collectively highlight the complexity and heterogeneity of promoter methylation patterns in B-ALL (Figure 1), which vary significantly across age groups, disease subtypes, and clinical stages. While adult and AYA B-ALL cases frequently exhibit promoter hypermethylation of tumor suppressor genes such as *CDKN1C*, *CDKN2A*, and *TP73*, pediatric cases often show transcriptional silencing through alternative, methylation-independent mechanisms. Subtype-specific methylation signatures, such as those observed in MLLr, *BCR::ABL1*, and *E2A::PBX1* B-ALL, further underscore the role of epigenetic regulation in leukemogenesis and relapse. Importantly, several of these methylation events are reversible and correlate with therapeutic resistance or prognosis, offering potential biomarkers for risk stratification and targets for epigenetic therapy.

#### 3.2.2. Genome-Wide Approaches

One of the first genome-wide studies in childhood B-ALL described a subgroup of patients harboring a relapse-associated signature characterized by hypermethylation of lymphocyte differentiation related genes (*FOXP1*, *TCF3*, *BLNK*, *CD79A*, *RAG1,* and *RAG2*) and hypomethylation of developmental genes (*HOX* genes and Polycomb target genes) [49]. The earliest large-scale study of DNA methylation in pediatric B-ALL analyzed 663 samples across major genetic subtypes. This study identified subtype-specific differentially methylated CpG sites and mapped their distribution across functional genomic regions. Importantly, it linked specific methylation changes to gene expression regulation and the risk of relapse. For instance, hypermethylation of *ERVH-3*, *C1orf222*, *KCNA3*, *PAG1*, *DNMBP*, and *C11orf52* were associated with relapse in *ETV6::RUNX1* cases, while *ZSCAN18*, *ZNF544*, *TAPBP::DAXX*, *WT1*, *ZNF681*, *ADARB2*, *ZNF329*, and *ZNF526* were implicated in MLLr cases. In *BCR::ABL1* patients, *LOC146880* hypermethylation was linked to relapse [50]. Building on this dataset, the same group later developed a highly sensitive subtype classification model based on 246 CpG sites, most of which were hypomethylated relative to non-leukemic controls. This classifier successfully assigned methylation subtypes to previously unclassified cases and, through RNA-seq validation, uncovered novel fusion genes such as *RUNX1::ASXL1*, *ETV6::CBX3*, *ETV6::AK125726*, *PAX5::ZCCHC7*, *PAX5::ETV6*, and the newly described PAX5/ESRRB and BRD9/NUTM1 fusions. These findings demonstrated the potential of DNA methylation profiling not only for subtype classification but also for uncovering cryptic genetic alterations [51].

Gabriel et al. further validated the correlation between genome-wide methylation profiles and cytogenetic subtypes such as dic(9;20), hyperdiploid, *TCF3::PBX1*, and *ETV6-::RUNX1* in a cohort of 52 pediatric B-ALL patients. While they confirmed the subtype-specific methylation signatures described by Nordlund et al., they did not observe any consistent methylation patterns associated with relapse, suggesting that relapse-related methylation changes may be more subtle or context-dependent [52]. Figueroa et al. also performed genome-wide methylation profiling in 137 pediatric B-ALL cases and found that methylation patterns clustered according to genetic subtype. In addition to subtype-specific changes, they identified a common methylation signature across B-ALL, consisting of 75 hypermethylated and nine hypomethylated regions. These regions included genes involved in cell cycle regulation (*MCTS1*, *DGK6*), RNA metabolism (*PABPN1*, *PABPC5*), signaling (*TIE1*, *MOS*, *CAMLG*, *GPRC5C*), transcriptional regulation (*PROP1*, *TAF3*, *H2AFY2*, *ELF5*, *ZBTB16*, *CNOT1*, *TADA2A*), and homeobox genes (*HOXA5*, *HOXA6*). They also confirmed the hypermethylation of known tumor suppressors such as *CDKN2A*, *CDKN2B*, and *PTEN*, and hypomethylation of the *KRAS* promoter [53]. A separate study analyzing 231 pediatric B-ALL cases revealed a dual epigenetic remodeling process. In both hyperdiploid and *ETV6::RUNX1* subtypes, *de novo* methylation of CpG islands in promoters and enhancers was observed, whereas large intergenic regions, particularly lamina-associated domains, underwent genome-wide demethylation. Polycomb target genes and *CTBP2* binding sites were among those hypermethylated, suggesting a disruption in developmental gene regulation and the persistence of stem cell-like epigenetic features in leukemic cells [54].

In infant MLLr B-ALL, a distinct epigenetic profile was observed [55]. Although global hypomethylation was a hallmark, MLL-AF4+ cases also exhibited unique promoter hypermethylation patterns. By filtering out methylation changes associated with normal B-cell development, the study identified specific alterations in regulatory regions such as enhancers and bivalent promoters. Notably, AP-1 (*FOS*/*JUN*) and *RUNX* family transcription factors were both hypomethylated and overexpressed in MLLr B-ALL. Functional assays demonstrated that AP-1 inhibition significantly impaired leukemic proliferation both in vitro and in vivo. These findings highlight the therapeutic potential of targeting transcription factor-driven epigenetic deregulation in aggressive pediatric leukemias [36].

Genome-wide studies in adults are scarce. MLLr B-ALL was also characterized by significant cytosine hypomethylation linked to MLL fusion protein binding and H3K79 dimethylation, which induced the transcriptional upregulation of oncogenic targets like FLT3 and BCL6. For BCR-ABL1-positive B-ALL, an aberrant cytosine methylation pattern was centered around a cytokine network, with hypomethylation and overexpression of IL2RA(CD25). This overexpression was strongly associated with a poor clinical outcome. In E2A-PBX1-positive B-ALL, aberrant DNA methylation was strongly linked to the direct binding of the E2A-PBX1 fusion protein, suggesting that the protein directly remodels the epigenome to create an aggressive B-ALL phenotype [36]. Integrating multi-omic data, Song et al. developed a new classification system called COMBAT, which stratified adult B-ALL patients into three distinct cohorts: COMBAT1 and COMBAT2, which were both characterized by a hypo-CpG island methylator phenotype (hypo-CIMP), and COMBAT3, which was distinguished by hypomethylation at enhancer regions and was associated with a CIMP [56].

This information is summarized in Table 2.

Genome-wide DNA methylation studies have shown that B-ALL subtypes exhibit distinct epigenetic patterns linked to genetic alterations and developmental origins (Figure 1). These profiles enhance subtype classification, reveal cryptic fusions, and suggest novel therapeutic targets. While integration with transcriptomic and genomic data is deepening our understanding of leukemogenesis, most studies to date have focused on pediatric cases. Expanding this research to adult B-ALL is crucial to uncover age-specific methylation patterns and enhance diagnostic and prognostic strategies across all age groups.

#### 3.2.3. Repetitive Elements

In addition to locus-specific methylation changes affecting tumor suppressors and oncogenes, global DNA methylation alterations also contribute to leukemogenesis. Transposable elements, which make up nearly half of the human genome, are typically silenced by methylation in normal cells. While hypomethylation of these elements is a hallmark of many solid tumors, leading to retrotransposition and genomic instability, early studies in leukemia revealed an opposite pattern. Bujko et al. were among the first to investigate methylation of repetitive elements in pediatric B-ALL, analyzing 32 samples. They focused on LINE1 and ALU sequences, which together represent about 30% of the genome and serve as proxies for global methylation levels. Surprisingly, both LINE1 and ALU elements were found to be hypermethylated in pediatric B-ALL compared to infant controls. Moreover, global methylation levels positively correlated with leukocyte counts [57]. This hypermethylation is thought to reflect the immature state of B-cell precursors, as hematopoietic progenitors naturally exhibit higher global methylation levels, which decrease during differentiation. The enrichment of methylation in B-ALL may thus result from the accumulation of undifferentiated blasts. However, whether this pattern holds true in adult B-ALL remains unclear, highlighting the need for comparative studies across age groups.

### 3.3. Histone Marks

Although histone mark deregulation has been extensively reviewed in pediatric B-ALL [58,59], similar comprehensive analyses are lacking for adult cases. This section highlights key studies on chromatin readers, writers, and remodelers across age groups (Figure 2).

#### 3.3.1. Epigenetic Writers

The MLLr subtype, driven by t(4;11), is a high-risk form of B-ALL that is significantly more prevalent in adults than in children. This rearrangement results in the loss of the H3K4 methyltransferase domain and fusion with *DOT1L*, leading to aberrant H3K79 methylation and transcriptional activation of leukemogenic genes [60]. In contrast, pediatric B-ALL more frequently harbors the *NSD2* p.E1099K variant, particularly in the *ETV6::RUNX1* and *TCF3::PBX1* subtypes. This mutation leads to increased H3K36me2 and supports leukemic growth both in vitro and in vivo. Additionally, this variant has not been detected in adult B-ALL, aligning with the rarity of these subtypes in older patients [61].

EZH2, a histone methyltransferase responsible for H3K27me3, has been studied in the AYA Nalm6 B-ALL cell line. It was found to silence tumor suppressors such as PTEN and p21. Silencing EZH2 or treating with Deazaneplanocin A reduced its expression, decreased proliferation, and induced apoptosis [62]. While this study focused on an AYA model, the role of EZH2 in adult B-ALL remains underexplored and warrants further investigation. YTHDC1, a reader of the m6A RNA modification, stabilizes KMT2C mRNA, enhancing H3K4me1/3 and promoting DNA repair. This mechanism contributes to B-ALL progression and drug resistance. Although the study did not specify patient age, both pediatric and adult cell lines were used, suggesting that YTHDC1 may be a relevant therapeutic target across age groups [63].

Histone acetylation patterns also show age-related differences. In AYA and adult B-ALL, high levels of H4 acetylation have been associated with favorable clinical outcomes, including improved survival and remission rates [64]. In contrast, pediatric B-ALL generally exhibits reduced H4 acetylation compared to non-leukemic cells. However, this mark is preserved in specific pediatric subtypes, such as those with *ETV6::RUNX1* fusion and *PAX5* deletion, which often co-occur and may reflect a distinct epigenetic profile [65]. HBO1, a histone acetyltransferase, is overexpressed in pediatric B-ALL and catalyzes H3K14ac, H4K8ac, and H4K12ac on the β-catenin (*CTNNB1*) promoter, activating the Wnt pathway. The inhibition of HBO1 with WM-3835 suppressed leukemic growth in vitro and improved outcomes in mouse models, highlighting its potential as a therapeutic target in pediatric B-ALL [66].

#### 3.3.2. Epigenetic Readers and Remodelers

BET (bromodomain and extra-terminal domain) proteins recognize acetylated lysins and mediate protein complex interactions. BRD4, the most studied family member, binds acetylated histones and recruits transcriptional activators to active chromatin. In pediatric B-ALL, Ott et al. first demonstrated the therapeutic potential of the BET inhibitor JQ1, which downregulated MYC and IL7R, disrupted the CRLF2–IL7R–JAK–STAT axis, and improved xenograft survival [67]. Da Costa et al. expanded on this by showing that JQ1 displaces BRD4 from the *MYC* promoter, leading to reduced MYC and IL7R expression, cell cycle arrest, apoptosis, and stalling of the replication fork. Notably, JQ1 selectively affected leukemic cells without harming normal peripheral blood mononuclear cells (PBMCs) and enhanced sensitivity to dexamethasone [68]. Furthermore, the BET degrader MZ1 was tested in both pediatric and AYA/adult B-ALL cell lines. MZ1 induces degradation of BRD2/3/4, resulting in CCND3 downregulation and disruption of the MYC, KRAS, and E2F signaling pathways. This leads to cell cycle arrest and apoptosis, suggesting that BET-targeting therapies may be effective across age groups, although in vivo validation is still needed [69].

Chromatin remodeling processes not only include histone PTMs, but also the activity of ATP-dependent complexes, such as SWI/SNF, that locates on promoters, gene bodies, and particularly enhancers to control transcriptional programs [70,71]. BRG1 (also known as SMARC4), a core subunit of the SWI/SNF complex, is overexpressed in adult B-ALL and promotes leukemogenesis by repressing *PPP2R1A*, activating PI3K/AKT signaling, and upregulating c-Myc. These effects were validated in both pediatric and adult cell lines, suggesting that BRG1 may serve as a therapeutic target across the age spectrum [72].

#### 3.3.3. Additional Histone Modifications Remarks

In pediatric B-ALL, Carroll et al. identified a relapse-specific gene expression signature characterized by upregulation of BIRC5, FOXM1, FANCD2, and TYMS, and downregulation of NR3C1, HRK, and SMEK2 [38]. Treatment with the HDAC inhibitor vorinostat increased H3K9ac and H3K4me2/3 at the promoters of *NR3C1* and *HRK*, reactivating their expression and suggesting a mechanism for reversing glucocorticoid resistance [39]. Although this study focused on pediatric models, the epigenetic regulation of *NR3C1* may have broader implications for treatment response in adult B-ALL as well. The commonly found pediatric B-ALL *IGH::DUX4* translocation was shown to preferentially target the epigenetically silenced IGH allele, characterized by hypermethylation and reduced H3K27ac. This results in moderate *DUX4* expression and reduced oncogenic stress, emphasizing the importance of epigenetic dosage control in leukemogenesis [73]. Finally, in Ph+ adult ALL, Van Dijk et al. observed that patients who did not respond to TKIs had lower levels of H3K4me2 and H3K4me3 at diagnosis, suggesting that these histone marks may serve as predictive biomarkers for treatment response in adult B-ALL [74].

In summary, histone modifications in B-ALL exhibit age-specific patterns: pediatric cases often display increased H3K36me2, reduced H4 acetylation, and EZH2-mediated repression, whereas adult cases more frequently exhibit KMT2A-driven H3K79 methylation and higher H4 acetylation, which is linked to a better prognosis. Shared alterations, such as BET protein activity and BRG1 overexpression, highlight common therapeutic targets, but further studies in adult B-ALL are needed to fully define its epigenetic landscape.

#### 3.3.4. Translational Gaps for Clinical Application of Histone PTMs-Related Epigenetic Therapies

As previously mentioned, extensive mechanistic work over the past decade has established the importance of aberrant histone modifications and dysregulated chromatin remodeling for B-ALL biology. In vitro and in vivo studies have shown that how BET inhibitors downregulate *MYC* and other pro-survival programs, impair replication fork progression, and enhance sensitivity to glucocorticoids and chemotherapy in resistant models (including *BCR:ABL1* and MLLr) [67,68]. Similarly, HDAC inhibitors such as vorinostat, panobinostat, romidepsin and newer class-selective agents restore the expression of epigenetically silenced tumor suppressors, reverse relapse-associated gene signatures, and resensitize leukemic blasts to multiple chemotherapeutics [39,75].

Despite the existing data supporting the use of BET and HDAC inhibitors in B-ALL, neither of them has achieved regulatory approval for pediatric or adult patients. Clinical translation has been hindered by several factors [76,77]. First, early-phase trials have demonstrated only modest and heterogeneous efficacy, with low complete remission rates and short response durations compared with robust preclinical activity. This may be related to the enrolment of broad hematologic malignancy cohorts in which B-ALL represents a small fraction, which limits the generation of disease-specific data. Moreover, the pronounced heterogeneity of B-ALL across genetic subtypes and age groups can attenuate the therapeutic impact observed in homogeneous in vitro models. Furthermore, tolerability issues restrict dosage or require intermittent schedules that may compromise efficacy. For that reason, durable benefit in this context generally requires rational combination of drugs, which are not yet defined for BET and HDAC inhibitors in the B-ALL scenario. Lastly, no validated predictive biomarkers exist to identify and stratify responders, which contributes to underwhelming trial outcomes in unselected patient populations.

In adults with B-ALL, these challenges are further magnified by age-specific barriers to epigenetic drug development. The lower incidence of the disease compared with pediatric cases, combined with substantial cohort fragmentation across molecular subtypes, limits the number and power of cases enrolling adult-only trials. Adult B-ALL exhibits greater biological heterogeneity, with higher frequencies of adverse cytogenetic and epigenetic lesions that may reduce uniform sensitivity to BET or HDAC inhibition. Comorbidities and reduced tolerance for intensive treatments limit the feasibility of integrating these agents for durable remissions. For this reason, combination strategies showing strong preclinical synergy frequently cannot be added to adult regimens without unacceptable toxicity. Moreover, most BET and HDAC inhibitor studies to date have either excluded adults or included them only as a minor subset within mixed hematologic malignancy cohorts, resulting in nonexistent adult-specific efficacy data to guide regulatory and clinical decision-making.

The progress for using BET and HDAC inhibitors in B-ALL will require biomarker-driven, age-specific trial designs that address genomic heterogeneity, comorbidities, and treatment tolerance. For adults, adaptive and specific trials could improve their outcomes. Preclinical work should focus on rational combinations with targeted or immunotherapies in realistic dosages, paired with pharmacodynamic monitoring and toxicity control, to translate mechanistic promises into effective, tolerable therapies for both pediatric and adult patients.

### 3.4. ncRNAs

Non-coding RNAs (ncRNAs) are a broad class of RNA molecules that do not encode proteins but act as key regulators of gene expression [78]. The main classes include microRNAs (miRNAs), typically around 22 nucleotides in length, which bind target mRNAs to induce degradation or block translation; long non-coding RNAs (lncRNAs), defined as transcripts longer than 200 nucleotides without coding potential, which can scaffold chromatin complexes by decoying other RNAs; and circular RNAs (circRNAs), covalently closed RNA molecules with stability in circulation [79,80,81]. These ncRNAs are now recognized as crucial players in cancer biology, including leukemogenesis [82].

Comprehensive descriptions of ncRNAs in ALL previously described in the literature [83,84,85,86,87,88] have revealed how both pediatric and adult B-ALL rely on ncRNA-mediated dysregulation of core leukemia-driving pathways, including PI3K/AKT/mTOR, JAK/STAT, NF-κB, cell cycle, and apoptosis. However, pediatric B-ALL showcases a broader and more extensively annotated spectrum of ncRNAs and their functional connection, including more detailed associations with differentiation, subtype- and relapse-specific biology, immune regulation, and epigenetic control, whereas the adult B-ALL literature is more limited in scope and most focused on Ph+ subtype, cell survival, and therapy response pathways, with less pathway diversity and fewer age-specific ncRNA discoveries to date [83,84,85,86,87,89,90,91,92,93].

Recent studies support the detection of blood-derived extracellular vesicle RNAs (containing ncRNAs and mRNA) as a powerful tool for the detection and monitoring of pediatric B-ALL, as those RNAs present disease-specific signatures and the sample acquisition is in a non-invasive manner compared to the regularly performed BM biopsies on B-ALL patients [94,95]. Although these are promising larger studies, technical refinement and longitudinal patient follow-up would be needed for clinical translation.

While pediatric B-ALL ncRNA profiling has yielded extensive subtype-specific signatures and functional validations, corresponding data in adults remain sparse (Table 3), often limited to secondary analyses or mixed-age cohorts, which impedes understanding of whether ncRNA networks are conserved, age-modulated, or uniquely dysregulated in older patients. Future investigations should prioritize age-stratified transcriptomic and epigenomic integration, longitudinal sampling across disease stages, and functional modelling in adult-derived systems. Such frameworks should aim to identify biomarkers predictive of therapy response, relapse, and survival in an age-appropriate clinical setting, enabling ncRNA-guided risk stratification and targeted interventions.

## 4. Concluding Remarks and Future Directions

While this review has primarily contrasted pediatric (<18 years) and adult (≥18 years) B-ALL, emerging genomic and epigenomic evidence indicates that the biology of the disease evolves along a continuum across the lifespan, rather than undergoing an abrupt transition at the 18-year threshold. Infants (<1 year) represent a distinct subgroup defined by a high prevalence of KMT2A (MLL) rearrangements, pro-B immunophenotype, and prenatal epigenetic programs that drive poor outcomes despite intensive therapy. In contrast, children between 1 and 14 years—particularly those aged 2 to 5—often present with favorable subtypes such as high-hyperdiploidy and *ETV6::RUNX1*, which are associated with characteristic methylation profiles and excellent survival. Adolescents and young adults (AYAs, ~15–39 years) occupy an intermediate position, with declining frequencies of favorable lesions and increasing prevalence of high-risk subtypes, including *BCR::ABL1*, Ph-like ALL, low hypodiploidy with *TP53* mutations, and *PAX5* alterations, yet they still retain certain “pediatric-like” molecular features. Beyond 40 years, unfavorable genomic and epigenomic alterations become increasingly dominant, including *BCR::ABL1* with *IKZF1* lesions and mutations in epigenetic regulators (e.g., *TET2*, *DNMT3A*, *IDH1/2*). In elderly patients (>70 years), the cumulative burden of adverse lesions, age-related clonal hematopoiesis, and limited therapeutic tolerance define a particularly high-risk group.

Across all ages, epigenetic dysregulation emerges as a key determinant of leukemic identity, transcriptional plasticity, immune evasion, and therapy response. Pediatric cases frequently harbor *CREBBP* and *SETD2* mutations, while adult patients more often present with alterations in *TET2*, *KMT2A*, and *IDH1/2*, pointing to age-specific axes of epigenetic deregulation. These findings underscore the urgent need for large-scale, age-stratified, multi-omic studies integrating genomics, methylation, histone modifications, chromatin accessibility, transcriptomics, proteomics, and metabolomics. Such approaches, particularly at the single-cell level, will clarify how age shapes clonal architecture, drives disease evolution, and influences relapse biology in the R/R setting.

The rapid evolution of genomic technologies has already transformed the molecular taxonomy of ALL, enabling classification of more than 90% of patients into genetically defined subgroups with significant clinical implications for diagnosis, risk stratification, and treatment. Yet, further research is required to define the pathogenic mechanisms of these abnormalities, their cooperative interactions, and their therapeutic vulnerabilities. Functional interrogation using CRISPR-Cas9, patient-derived xenografts, and organoid models will be indispensable for establishing causal roles and uncovering druggable dependencies.

While chemotherapy remains the backbone of treatment, there is a pressing need to expand targeted and immune-based therapies. Next-generation strategies include:

Targeted inhibitors: Efforts to optimize CDK4/CDK6 blockade (e.g., combination regimens or PROTACs), PI3K/AKT/mTOR inhibition (rapalogues in trials), BH3 mimetics such as venetoclax, and menin inhibitors for *KMT2A*-rearranged leukemias, several of which have shown promising clinical activity.

Subtype-specific therapies: FLT3 inhibitors for *ZNF384*-rearranged B-ALL and HDAC inhibitors for MEF2D-rearranged disease represent emerging avenues.

Immunotherapies: Advances in monoclonal antibodies, bispecific T-cell engagers (BiTEs), antibody–drug conjugates (ADCs), and CAR-T cells—including allogeneic and next-generation constructs—are likely to expand therapeutic options.

Looking ahead, the field should prioritize the following research questions:

How do genetic and epigenetic programs evolve across age groups, and which features are true drivers of prognosis and treatment response?What mechanisms underlie clonal heterogeneity and relapse biology, and how can they be therapeutically intercepted?Can age-adapted, multi-omic biomarkers improve early diagnosis, risk stratification, and MRD monitoring in a minimally invasive manner?How can rational combinations of epigenetic drugs, targeted inhibitors, chemotherapy, and immunotherapy be optimized for AYAs and elderly patients?What strategies will ensure global accessibility and equitable implementation of precision-based therapies?

In conclusion, B-ALL represents a dynamic spectrum of age-dependent diseases shaped by the interplay of genetic and epigenetic mechanisms. The integration of cutting-edge multi-omics, precise functional validation, and rational drug development—together with immunotherapy and novel targeted approaches—holds the promise of globally accessible, personalized, and synergistic treatment strategies. Achieving this goal will require collaborative efforts to translate molecular insights into standardized clinical practice, ultimately improving survival and quality of life for B-ALL patients of all ages.

## Figures and Tables

**Figure 1 ijms-26-08774-f001:**
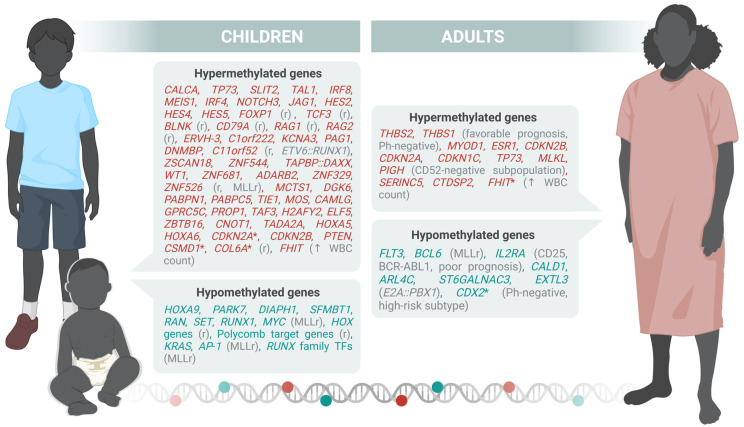
Differentially methylated genes in children and adults. Those described also for AYA are indicated with the symbol *. The letter “r” refers to genes associated with relapse. Hypermethylated genes are colored in red and hypomethylated genes are colored in blue. Illustration generated with Biorender.com.

**Figure 2 ijms-26-08774-f002:**
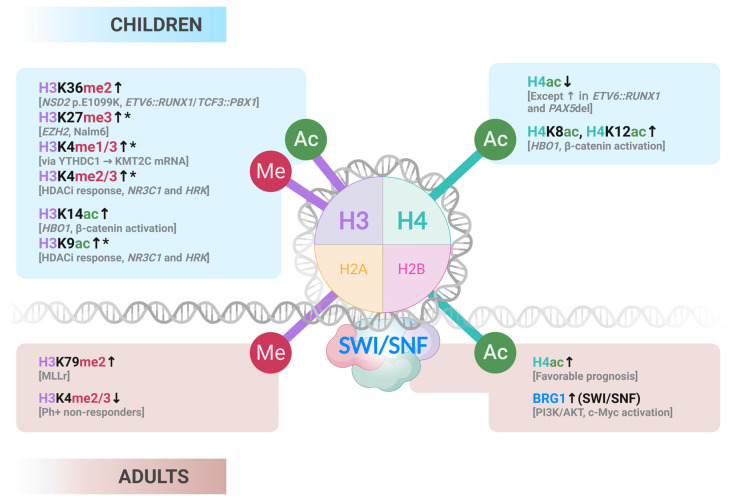
Described histone modifications found in children and adults. Those described also for AYA are indicated with the symbol *. Arrows indicate if the mark is increased (↑) or decreased (↓). Illustration generated with Biorender.com.

**Table 1 ijms-26-08774-t001:** Genetics and prevalence of B-ALL subtypes in pediatric and adult populations.

Category	Subtype	Pediatric Frequency	Adult Frequency	Definition/Characteristics	Impact
Chromosomal Aneuploidy	High hyperdiploidy	~30%	<10%	Gain of ≥5 chromosomes; nonrandom gains (e.g., 4, 10, 14, 21, X). Often *RAS* and epigenetic mutations.	Good in children, favorable in adults.
	Low hypodiploidy	<1%	5–15%	32–39 chromosomes. *TP53* mutations, germline in children, somatic in adults.	Poor, increasing with age.
	Near haploidy	∼2%	<1%	24–31 chromosomes. *RAS* mutations, *IKZF3* deletions.	Very poor prognosis.
Chromosomal Abnormalities	*BCR::ABL1* (Ph+)	2–5%	20–40%	t(9;22). Common in older adults. *IKZF1* deletions frequent.	Historically poor; better with TKIs.
	*KMT2A*-rearranged (MLLr)	∼80% in infants	5–15%	t(v;11q23), often *AFF1*. High WBC, therapy-related.	Poor; needs intensive or novel treatment.
	*ETV6::RUNX1*	∼25%	1–2%	t(12;21). Frequent in pediatric B-ALL.	Good in children; favorable in AYA.
	*TCF3::PBX1*	2–6%	2–6%	t(1;19). CNS involvement common.	Prognosis variable; CNS-directed therapy important.
	iAMP21	∼3%	<2%	*RUNX1* amplification. Rare in adults.	Poor unless intensively treated.
	HLF-rearranged	<1%	<1%	t(17;19); t(17;18). Rare, associated with hypercalcemia.	Dismal prognosis.
	*PAX5*alt	∼10%	5–10%	Multiple alterations in *PAX5*; frequent *CDKN2A*, *IKZF1* deletions.	Intermediate to poor prognosis.
	*DUX4*-rearranged	∼8%	2–10%	t(4;14). CD371+, *IGH* rearranged, *IKZF1* and *ERG* deletions.	Favorable prognosis.
Point Mutations	*PAX5* P80R	<2%	3–6%	Missense mutation in *PAX5*; JAK-STAT/RAS pathway mutations common.	Favorable to intermediate.
	*IKZF1* N159Y	<1%	<1%	Rare point mutation with Chr 21 gain.	Unknown significance.
	*IDH1/2*	ND	1–2%	Metabolic gene mutations.	Poor prognosis.
Transcriptional Signatures	*BCR::ABL1*-like (Ph-like)	10–15%	15–30%	*CRLF2* rearranged or kinase-activated; resembles Ph+ without *BCR::ABL1* fusion.	Poor; may respond to TKIs and targeted therapies.
	*ETV6::RUNX1*-like	∼3%	<1%	Similar expression to *ETV6::RUNX1*; *ARPP21*, *IKZF1* deletions.	Likely favorable.
	*KMT2A*-like	<1%	<1%		Poor prognosis.
	*ZNF384*-like	∼5%	2–8%	Mimics *ZNF384*-r with kinase and epigenetic pathway alterations.	Intermediate prognosis.

**Table 2 ijms-26-08774-t002:** DNA methylation genome-wide studies’ specifications.

Reference	Population	Lineage	Altered Genes
Sandoval et al., 2012 [49]	29 pediatric B-ALL (normal karyotype, hyperdiploid, pseudodiploid, others; relapse vs. non-relapse)	B	**Hypermethylated** (relapse group): FOXP1, TCF3, BLNK, CD79A, RAG1, RAG2. **Hypomethylated**: HOXA cluster (HOXA5/6/9), Polycomb target genes
Nordlund et al., 2013 [50]	764 pediatric ALL (663 B, 101 T) + 27 relapse samples	B + T	**Core hypermethylated (9406 CpGs):** CDKN2A, CDKN2B, PTEN, TP73, DAPK1, WIF1, SFRP2/5, APC, HOXA5/6/9, TIE1, MOS, PCDH loci. **Subtypes**: HeH (**hypomethylated**: DDIT4L, PTPRG, FHIT); MLL-r (**hypermethylated**: BNIP3, ZAP70, XYLT2, HLA-B, EDEM1); Relapse (**hypermethylated**: CDH3, TBX2, ERCC1, NPR2, DAPK1, CCR6, HRK, LIFR1, DLX3)
Nordlund et al., 2015 [51]	546 pediatric ALL (7 B-ALL subtypes, T-ALL)	B + T	ETV6-RUNX1: **Hypermethylated** promoters: CDKN2A/B, PTEN, DAPK1. MLL-r: HOXA cluster (HOXA9-11) **hypomethylated**; PcG targets **hypermethylated**. Classifier CpGs included EPOR, ASNS, TCF3, PBX1, and EBF1 regulatory regions
Gabriel et al., 2015 [52]	52 pediatric B-ALL (ETV6-RUNX1, HeH, TCF3-PBX1, dic(9;20))	B	ETV6-RUNX1 and dic(9;20): mostly **hypermethylated** (E2F6, DCC, NKX6-1, PTPN6). HeH and TCF3-PBX1: broad **hypomethylation** (ZNF clusters, intergenic). No consistent relapse-specific DMCs
Figueroa et al., 2013 [53]	167 pediatric ALL (137 B, 30 T)	B + T	**Hypermethylated** (66 genes): TIE1, MOS, CAMLG, GPRC5C, MCTS1, DGKG, PABPN1, PABPC5, PROP1, TAF3, H2AFY2, ELF5, ZBTB16, CNOT1, TADA2A, HOXA5/6, CDKN2A/B, PTEN, BNIP3, DAPK1, SYK, BRINP1, WIF1. **Hypomethylated**: KRAS, FUT9, ADCY2, CETN3, ELF5, DNTT
Lee et al., 2015 [54]	227 pediatric B-ALL (ETV6-RUNX1, HeH, others)	B	ETV6-RUNX1: **hypermethylated** ASNS, EPOR, PDK4, SYT family. HeH: global intergenic demethylation. PcG targets (GATA4, HLF, PAX5/6, HOXD1) **hypermethylated**
Tejedor et al., 2021 [55]	69 infant B-ALL (37 MLL-AF4+, 12 MLL-AF9+, 20 non-MLLr)	B	**Hypomethylated**: FOS, JUN, RUNX1 (AP-1 network). **Hypermethylated**: developmental PcG targets (SOX2, OCT4, NANOG genes, DAPK1, CCR6, HRK, LIFR1, FHIT)
Geng et al., 2012 [36]	215 adult B-ALL patients enrolled in a phase III clinical trial (ECOG E2993).	B	**Hypomethylated**: FLT3 and BCL6, IL2RA(CD25).
Song et al., 2025 [56]	88 adult B-ALL patients (69 newly diagnosed and 19 relapsed/refractory).	B	**Hypomethylated**: MYC

**Table 3 ijms-26-08774-t003:** Classification by age group of ncRNAs identified to be altered in B-ALL.

Age Group	Type	Affection in B-ALL	ncRNAs
Pediatric	miRNA	Upregulation	miR-155, miR-181a, miR-128, miR-130b, miR-210, miR-222, miR-708, miR-363
		Downregulation	miR-125b, miR-143, miR-148a, miR-223, miR-145, let-7e, miR-100, miR-340, miR-335
	lncRNA	Upregulation	AWPPH, BALR-2, CRNDE, MALAT1, LINC00958, RP11-252C15.1, ZEB1-AS1, DUXAP8
		Downregulation	LINC00926, AC009495.3, CECR7, RP11-624C23.1, AC083949.1, SNHG16
	circRNA	Upregulation	circAF4
	EV-associated ncRNA	Disease discrimination	let-7f-5p, miR-26b-5p, miR-335-5p (down in B-ALL); miR-4645 (up in B-ALL EVs)
Adult	miRNA	Downregulation	miR-183-5p (in Ph+ B-ALL); miR-29a (drug resistance context); miR-125b (mixed data, often lower in adults)
	lncRNA	Upregulation	DUXAP8 (chemoresistance), ZEB1-AS1 (STAT3 activation, functional data appear in adults as well as pediatric)
Both	miRNA	Both	miR-29a (drug resistance), miR-125b (over/under expressed by subgroup), miR-146a (often up, subtype- and context-specific)

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
