# Peer review of "Age-Driven Genetic and Epigenetic Heterogeneity in B-ALL"

_ijms, 2025, doi:10.3390/ijms26188774_

Round 1

Reviewer 1 Report

Comments and Suggestions for Authors

The review paper,analysed B-ALLas a very heterogenous disease, both inthe clinical and biological sense. It very systematicali and comprehensively compares the characteristics of the disease depending on age: infants/young adults/adluts. The autors uses current literature in the analysis of genetic and epigenetic changes in B ALL, with clear explanantion of the impact of those changes on the prognosis. Additionaly, the connection of various molecular changes in B ALL with the applied therapyis explained in detail, for example the actionof hypomethylation agents in patients with hypomethylation processes. Special compliments for the tabular and shematic presentation of  molecular changes depending of the age of B ALL patients.  

Author Response

Comment: The review paper,analysed B-ALLas a very heterogenous disease, both inthe clinical and biological sense. It very systematicali and comprehensively compares the characteristics of the disease depending on age: infants/young adults/adluts. The autors uses current literature in the analysis of genetic and epigenetic changes in B ALL, with clear explanantion of the impact of those changes on the prognosis. Additionaly, the connection of various molecular changes in B ALL with the applied therapyis explained in detail, for example the actionof hypomethylation agents in patients with hypomethylation processes. Special compliments for the tabular and shematic presentation of  molecular changes depending of the age of B ALL patients.  

Response: We sincerely thank the reviewer for the positive evaluation and encouraging feedback. We are pleased that the systematic approach, comprehensive analysis, and the integration of schematic and tabular presentations were found to be clear and informative.

Reviewer 2 Report

Comments and Suggestions for Authors

In the work, „A Tale of Two Leukemias: Age-Driven Genetic and Epigenetic Heterogeneity in B-ALL”, the authors Veselinova et al. present a nice and informative overview on an important field; the work is well written and presents many interesting aspects of knowledge. In particular, it gives a concise yet exhaustive overview on numerous genetic and in particular epigenetic factors regulating pediatric and adult B-ALL, with high prognostic implications and a possible role future therapeutic approaches.

To me, it is only one aspect that still must be considered.

In the paper, there basically is only a discrimination between pediatric and adult leukemias; other age subgroups are not mentioned in detail. In fact, it is well known that genetic subtypes of infants are different from those of older children. Further, AYA patients – generally defined as the age group between 15th  and 40th  birthday – are mentioned in the text, however, detailed information on more specific age groups  are lacking. Is the genetic profile of a 20 years old identical to that of an 80 years old patient? Is it really the 18th birthday which discriminated best between pediatric and adult subtype? Are patients between 15th and 18th birthday belonging to the pediatric or the adult subgroup? Such aspects may be different in the papers on which this review is based, and should be mentioned in the discussion.

Elsewise, I think the work is perfect.

Author Response

To me, it is only one aspect that still must be considered.

In the paper, there basically is only a discrimination between pediatric and adult leukemias; other age subgroups are not mentioned in detail. In fact, it is well known that genetic subtypes of infants are different from those of older children. Further, AYA patients – generally defined as the age group between 15th and 40th birthday – are mentioned in the text, however, detailed information on more specific age groups  are lacking. Is the genetic profile of a 20 years old identical to that of an 80 years old patient? Is it really the 18th birthday which discriminated best between pediatric and adult subtype? Are patients between 15th and 18th birthday belonging to the pediatric or the adult subgroup? Such aspects may be different in the papers on which this review is based, and should be mentioned in the discussion.

Response: We thank the reviewer for the insightful comments and constructive suggestions. We fully agree that the molecular landscape of B-ALL varies across age groups and cannot be strictly dichotomized into pediatric versus adult subtypes. In response to this valuable feedback, we have modified the title of the manuscript and emphasized throughout the text—particularly in the discussion—that the age continuum shapes the genetic and epigenetic features of B-ALL. We now highlight the specificities of infants, adolescents/young adults (AYA), and elderly patients, while acknowledging that some of these differences remain underexplored in the current literature. But to provide a broad yet concrete overview, we still have primarily structured our synthesis around the pediatric/adult dichotomy, as most available data are organized in this manner. We believe this balance allows us to capture the most robust findings while recognizing the nuances and limitations in age-specific subgrouping.

Reviewer 3 Report

Comments and Suggestions for Authors

Veselinova et al. submitted an interesting review article that presents a comprehensive review of the epigenetic mechanisms underlying B-ALL, focusing on age-related differences between pediatric, AYA, and adult cases. The manuscript is a valuable and timely review that integrates a large body of recent literature on epigenetics in B-ALL. However, I have some comments/suggestions that might be taken into consideration prior to the publication, and I make them in section-by-section as follows: 
1. Introduction: I believe there is a limited discussion of why epigenetic approaches have been slower to develop in adult B-ALL compared to pediatric cases. Also, authors should provide a clearer statement of purpose and add a short summary sentence on how this review adds novelty compared to previous ones.
2. DNA methylation and genome-wide studies: Would it be possible to add a comparative table summarizing methylation alterations by age group and subtype? In addition to this comment, I have a concern whether the studies cited in adults are based on sufficient sample sizes to make definitive conclusions.
3. Histone modifications and chromatin remodeling: Could you include a subsection discussing translational gaps between mechanistic studies and clinical trials and please expand on possible reasons for limited application of BET inhibitors and HDAC inhibitors in adults?
4. Non-coding RNAs: I feel that this section is disproportionately focused on pediatric data, while adult-specific findings are limited to brief mentions. Thus, I highly recommend that authors should first emphasize gaps in adult ncRNA studies, and propose a framework for future age-stratified investigations. Second, add a concise table categorizing key ncRNAs and their functional implications by age group.
5. Conclusion and future directions: I highly encourage authors to provide specific, prioritized research questions that the field should address and it is essential to suggest how emerging multi-omic approaches can bridge the pediatric-adult knowledge gap.

Author Response

I have some comments/suggestions that might be taken into consideration prior to the publication, and I make them in section-by-section as follows: 

  1. Introduction: I believe there is a limited discussion of why epigenetic approaches have been slower to develop in adult B-ALL compared to pediatric cases. Also, authors should provide a clearer statement of purpose and add a short summary sentence on how this review adds novelty compared to previous ones.

Response: We thank the reviewer for this valuable suggestion. We have expanded the introduction to include a discussion on the slower progress of epigenetic studies in adult B-ALL, emphasizing factors such as lower incidence, smaller available cohorts, and limited clinical trial enrollment compared to pediatric populations. In addition, we have clarified the statement of purpose highlighting age-dependent mechanisms and translational implications not comprehensively addressed in previous reviews.

  1. DNA methylation and genome-wide studies: Would it be possible to add a comparative table summarizing methylation alterations by age group and subtype? In addition to this comment, I have a concern whether the studies cited in adults are based on sufficient sample sizes to make definitive conclusions.

Response: We fully agree with the reviewer that adult-focused studies often rely on smaller sample sizes, which limits the strength of definitive conclusions. We have emphasized this limitation in the text. Following the reviewer’s recommendation, we have also incorporated Table 2, which summarizes genome-wide methylation alterations across age groups and subtypes. In addition, we now provide a detailed table of targeted gene methylation studies to offer a more complete comparative overview.

Please note below a table of the studies included from targeted gene studies, indicating the number of patients.

Reference

Population (Age/Subtype)

Lineage

Altered Genes and specifications

Leegwater et al., 1997 [29]

14 pediatric ALL

B + T

Hypermethylated: CALCA promoter (13/14 at diagnosis). Longitudinal: further methylation changes at relapse. T‑ALL > B‑ALL methylation (statistically significant).

Corn et al., 1999 [30]

35 pediatric ALL + 10 Burkitt’s lymphoma cases; leukemia cell lines (U937, HL‑60).

B + T

Hypermethylated: p73 promoter (T‑ALL ~62% methylated vs B‑ALL ~17%). Associated with transcriptional silencing; restored by 5‑aza‑dC.

Garcia‑Manero et al., 2002 [31]

80 untreated adult ALL

B + T

Hypermethylated: MDR1 (ABCB1) (mean 24.5%), THBS2 (20.8%), MYF3 (17.6%), ER (16.1%), p15/CDKN2B (11.3%), THBS1 (8.9%), CD10 (4.5%), ABL1 (3.7%), p16/CDKN2A (1.3%), p73 (21.2%). Multi‑gene methylation frequent (86% ≥1 gene).

Shen et al., 2003 [32]

72 adult ALL (pretreatment), 21 with paired relapse samples

B + T

Hypermethylated: p57KIP2 (CDKN1C). Combined p73+p15+p57 hypermethylation in 22% Ph‑negative patients, with transcriptional silencing and poor survival.

Bueso‑Ramos et al., 2005 [33]

64 adult ALL

B + T

Triad of p73, p15, p57Kip2 frequently methylated. Low protein levels predict poor outcome, independent of methylation.

Gutiérrez et al., 2005 [34]

74 childhood ALL; adults from [32]

B + T

Hypermethylated: Childhood ALL: p73 (~20%), p15 (~20%), p57 (7%). Adults: p73 (~22%), p15 (~20%), p57 (~50%). p73 expression loss is very common in children (alternative epigenetic silencing).

Stumpel et al., 2011 [35]

15 infants with MLLr ALL, 10 without MLLr

B

Distinct genome‑wide hypermethylation + focal hypomethylation of proto‑oncogenes (MYC, HOXA9, SET, RUNX1, RAN, PARK7, DIAPH1). HDAC inhibitors reverted hypomethylated oncogene profiles.

Geng et al., 2012 – [36]

215 adult B‑ALL, subtype-specific:

BCR–ABL1+ (n=83)

E2A–PBX1+ (n=7)

MLL‑rearranged (n=28).

B

BCR–ABL1+: IL2RA (CD25) hypomethylated and overexpressed; linked to chemoresistance.

E2A–PBX1+: Direct fusion‑dependent remodeling, hypomethylation at specific targets.

MLL‑r: Broad hypomethylation, H3K79me2 marks, upregulation of BCL6, FLT3.

Dunwell et al., 2009 [37]

64 pediatric ALL (52 B-ALL, 12 T-ALL); 30 adult CLL; 10 leukemia cell lines

B + T

Hypermethylation of tumor suppressor gene SLIT2 promoter (58% B‑ALL; 83% T‑ALL; 80% CLL). Absent in normal BM/lymphocytes. Silencing reversible by 5‑aza‑dC.

Hogan et al., 2011 [38]

61 pediatric ALL (55 B-ALL, 6 T-ALL), matched diagnosis/relapse

B + T

Identified 984 relapse‑specific methylated genes: 251 hypermethylated + silenced (CDKN2A, COL6A2, PTPRO, CSMD1), 29 hypomethylated + upregulated (NOTCH4, TOP1MT, SLC7A7)

Relapse genome more hypermethylated; epigenetic silencing contributes to chemoresistance; some genes reactivated by hypomethylating agents.

Bhatla et al., 2012 [39]

Childhood B‑ALL, matched diagnosis/relapse, from [38], 33 paired samples with methylation data

B

Hypermethylated ~1147 CpGs/905 genes including CDKN2A, COL6A2, PTPRO, CSMD1.

Hypomethylation of ~81 CpGs/79 genes, including NOTCH4, TOP1MT.

Musialik et al., 2015 [40]

38 pediatric B‑ALL; 20 controls

B

Hypermethylated: TAL1 (26.3%), IRF8 (7.9%), MEIS1 (5.3%), HOXA4 (16%), IRF4 (2.6%) and HOXA5 (5%); the first 4 of them showing silencing at protein levels.

Kuang et al., 2013 [41]

54 B‑ALL, 14 T‑ALL patients; ALL cell lines. Healthy B cell controls included.

B + T

Hypermethylated in B‑ALL.

More in B-ALL than T-ALL: NOTCH3, HES5, HES4 (~70%).

Equally in B-ALL and T-ALL: HES2 (33% in B-ALL)
Hypermethylation correlates with gene silencing; reversible by DAC.

Hypomethylated: NOTCH1, NOTCH2, DLL1, DLL3, DLL4, HES6.

Abdullah et al., 2017 [42]

56 pediatric B‑ALL (42 chemoresponsive, 14 resistant), 6 healthy controls

B

Hypermethylated: ADAMTSL5 (93% resistant vs. 38% sensitive), CDH11 (79% vs. 40%). Not methylated in normal controls, linked to therapy resistance.

Loeff et al., 2019 [46]

25 pediatric + adult B‑ALL

B

Epigenetic silencing of PIGH (loss of GPI‑anchor expression) by promoter hypermethylation → CD52/CD55/CD59 deficiency, reversible by 5‑aza‑dC.

Ghantous et al., 2024 [47]

Pediatric ALL (663 B-ALL, 101 T-ALL) + 483 controls

B

Hypermethylation of VTRNA2‑1 (maternal imprinting gene), detectable at birth and consistent across tissues; prognostic for survival, proposed as leukemia precursor biomarker.

Mohammad et al., 2024 [48]

66 ALL cases (35 B-ALL, 31 T-ALL) + 66 healthy controls

B + T

Hypermethylated: FHIT promoter (significantly higher in ALL vs controls). Correlates with reduced FHIT expression. Downregulation linked to higher WBC count and poorer OS.

Yasuda et al., 2022 [28]

354 Ph- AYA and adult B‑ALL cases

B

Subtype classification of cases:

·  ZNF384‑rearranged (22.6%) most frequent.

·  Novel subtype CDX2‑high (3.4%) defined by high expression and chr1q gain (associated with poor prognosis).

·  Novel subtype IDH1/2‑mutant (1.9%) defined by IDH1 R132C or IDH2 R140Q mutations, hypermethylation profile, and poor prognosis.

  1. Histone modifications and chromatin remodeling: Could you include a subsection discussing translational gaps between mechanistic studies and clinical trials and please expand on possible reasons for limited application of BET inhibitors and HDAC inhibitors in adults?

Response: We thank the reviewer for this insightful comment. A new subsection has been added to highlight the translational gaps between mechanistic discoveries and clinical implementation. In particular, we discuss the reasons for the limited application of BET and HDAC inhibitors in adult B-ALL, including issues of trial design, toxicity profiles, and lack of large-scale, age-stratified efficacy data.

  1. Non-coding RNAs: I feel that this section is disproportionately focused on pediatric data, while adult-specific findings are limited to brief mentions. Thus, I highly recommend that authors should first emphasize gaps in adult ncRNA studies, and propose a framework for future age-stratified investigations. Second, add a concise table categorizing key ncRNAs and their functional implications by age group.

Response: We appreciate the reviewer’s constructive suggestion. We have revised this section to explicitly highlight the paucity of adult-specific data, framing it as a major gap in the field. To address this, we now propose a framework for future age-stratified investigations of ncRNAs. In addition, we have included Table 3, which categorizes key ncRNAs and summarizes their functional implications by age group

  1. Conclusion and future directions: I highly encourage authors to provide specific, prioritized research questions that the field should address and it is essential to suggest how emerging multi-omic approaches can bridge the pediatric-adult knowledge gap.

Response: We are grateful for this recommendation. The conclusion has been substantially revised to provide a clearer perspective, now including specific, prioritized research questions. We also emphasize how emerging single-cell and multi-omic approaches can be leveraged to bridge the pediatric–adult knowledge gap, with a particular focus on age-stratified biology, clonal evolution, and therapeutic vulnerabilities.

Round 2

Reviewer 3 Report

Comments and Suggestions for Authors

All comments have been addressed properly.